# Current Trends in Syphilis Mortality in the United States, 2015–2020

Noel C. Barragan [1], Ranjana N. Wickramasekaran [2], Frank Sorvillo [2], Lisa V. Smith [2] and Tony Kuo [2,3,4,*]

1   Division of Chronic Disease and Injury Prevention, Los Angeles County Department of Public Health, Los Angeles, CA 90010, USA
2   Department of Epidemiology, University of California, Los Angeles (UCLA) Fielding School of Public Health, Los Angeles, CA 90095, USA
3   Department of Family Medicine, David Geffen School of Medicine at UCLA, Los Angeles, CA 90024, USA
4   Population Health Program, UCLA Clinical and Translational Science Institute, Los Angeles, CA 90095, USA
*   Correspondence: tkuo@mednet.ucla.edu

**Abstract:** Rates of reported cases of syphilis have steadily increased since 2000 in the United States. However, despite the increase in cases, mortality from 2000–2014 declined. The following study examines the latest trends in syphilis-related deaths using 2015–2020 Multiple Cause of Death data. A total of 925 syphilis-related deaths were identified during the study period, 30% of which listed syphilis as the underlying cause of death. On average, age-adjusted syphilis mortality increased by 9.51% annually (95% CI = 5.41%–13.77%). Study findings indicate a marked increase in deaths attributed to syphilis, underscoring the need to more systematically and comprehensively address the growing sexually transmitted infection epidemic in the United States.

**Keywords:** syphilis; sexually transmitted infection; mortality

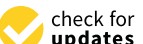



## 1. Introduction

The rates of mandatory reportable sexually transmitted infections, including syphilis, have reached epidemic proportions in the United States (US) and continue to rise [1]. Cases of syphilis have increased almost every year since hitting a historic low in 2000 (11.2 cases per 100,000 population), reaching 40.8 cases per 100,000 population in 2020 [2]. However, rates of disease have not affected all groups equally. For example, the rates of primary and secondary syphilis are highest among those aged 20–34, Blacks/African Americans, American Indian or Alaska Natives, and men (likely attributed to increases in cases among men who have sex with other men). Notably, the number of cases of congenital syphilis have also shown a marked increase over the last decade; from 2011 to 2020, cases increased 500% from 358 to 2148. The resurgence of syphilis parallels that of other sexually transmitted infections. For example, overall rates of reported cases of gonorrhea have increased 111% since 2009; data from 2016–2020 also consistently show Blacks/African Americans and American Indian or Alaska Natives as having the highest rates of reported cases. The unabated and unequal increases in syphilis and other sexually transmitted infections have been attributed to a number of social and contextual factors, including the following: sexual behavior trends, such as having multiple sex partners or practicing unsafe sex; human immunodeficiency virus (HIV) coinfection; substance use; limited access to or utilization of health care among certain populations who are more likely to be uninsured, lack transportation or childcare, face language barriers, lack trust in health care providers, or experience racism and discrimination; and an inadequate public health infrastructure [1,3,4]. Despite these consistently increasing trends in morbidity, syphilis mortality rates have not proven as discouraging. An analysis of syphilis-related mortality in the United States between 2000–2014 identified a total of 1829 deaths, 32% of which listed syphilis as the underlying cause of death [5]. During this time period, syphilis-related

mortality showed an overall average annual decline of −2.90% (95% Confidence Interval (CI), −3.39% to −1.87%). When disaggregated, declines were seen in nearly all subgroups, with Blacks/African Americans experiencing the largest statistically significant reduction (−5.19%; 95% CI, −6.63% to −3.72%). These contrasting morbidity and mortality trends are likely due to early disease detection and effective treatments; however, it is unclear if these pattens in the identification and management of the infection have continued. The present study serves as an update to this previous analysis, describing syphilis mortality rates in the United States from 2015–2020.

## 2. Materials and Methods

Syphilis mortality was examined using national-level, Multiple Cause of Death data obtained from CDC WONDER for the time period 2015–2020 [6]. All decedents with ICD-10 codes A50 (congenital syphilis), A51 (early syphilis), A52 (late syphilis), or A53 (other or unspecified syphilis) listed as the underlying or associated cause of death were included in the analysis. Mortality rates per 100,000 population were generated using bridged race population estimates provided by the National Center for Health Statistics. Age-adjusted mortality rates and rate ratios were calculated based on the 2000 US standard population in alignment with current US federal reporting practices [7]. Mortality trends over time were modeled using Poisson regression analysis. Given the known relationship between syphilis and HIV, a comparison of comorbid syphilis and HIV mortality was also conducted. HIV was identified using ICD-10 codes B20–B24 and R75.

All statistical analyses were performed using SAS, version 9.4 (SAS Institute, Inc., Cary, NC, USA). Due to the nature of the data (i.e., publicly available, not individually identifiable, and containing no information that can be linked to live human subjects), the present study did not require formal review or approval from an institutional review board.

## 3. Results

During the six-year study period, a total of 925 syphilis-related deaths were identified. Of these, 30% (n = 277) indicated syphilis as the underlying cause of death (Figure 1). A total of 52 deaths were attributed to congenital syphilis. The majority of deaths were in males (n = 612), who were more than twice as likely to die of syphilis than females (age-adjusted rate ratio (AARR) = 2.33; 95% CI = 2.06–2.64) (Table 1). Among race/ethnicity groups, while the highest number of deaths were among Whites (n = 423), Blacks/African Americans experienced the highest rates of syphilis deaths (age-adjusted mortality rate (AAMR) = 0.15 per 100,000 population; 95% CI = 0.14–0.17) and were more than five times as likely to die of syphilis compared to Whites (AARR = 5.86; 95% CI = 5.30–6.49). American Indian or Alaskan Natives also exhibited an increased risk of syphilis death compared to Whites. However, these values should be interpreted with caution due to statistical instability as a result of the small sample size. Decedents aged 85 years and older exhibited the highest death rate (age-specific mortality rate (ASMR) = 0.38 per 100,000 population; 95% CI = 0.31–0.44), followed by those aged 75–84 and those under 1 year of age, who both had an ASMR of 0.18 per 100,000 population (95% CI = 0.15–0.21 and 95% CI = 0.12–0.23, respectively). Overall, age-adjusted syphilis mortality increased by 9.51% annually (95% CI = 5.41%–13.77%). When examined by subgroup, those diagnosed with congenital syphilis exhibited the highest statistically significant annual increase (28.28%; 95% CI = 8.50%–51.68%) (Figure 2). Approximately 13% (n = 121) of decedents had HIV listed as a comorbid condition. The majority of those with comorbid syphilis and HIV were male (n = 112), and nearly 50% (n = 60) were Black/African American.

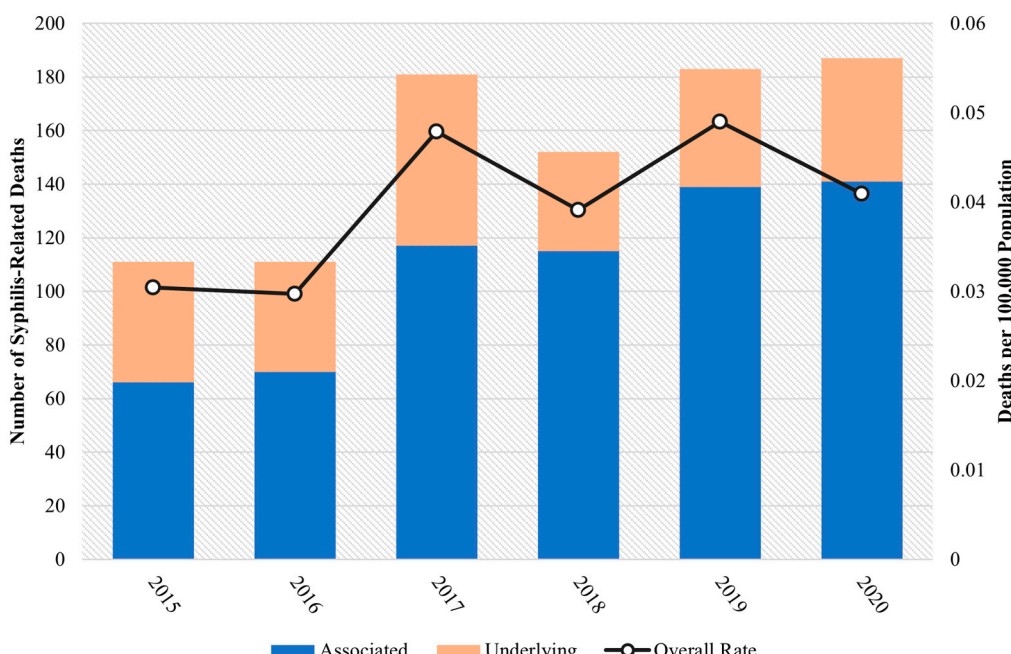

**Figure 1.** Number of underlying and associated syphilis deaths and overall age-adjusted rates per 100,000 population by year, Unites States, 2015–2020.

**Table 1.** Syphilis mortality rates and rate ratios per 100,000 population in the United States, 2015–2020.

| | Frequency n = 925 (%) * | Age-Adjusted Mortality Rate (95% CI) | Age-Adjusted Rate Ratio (95% CI) |
|---|---|---|---|
| **Sex** | | | |
| Female | 313 (33.8) | 0.03 (0.02–0.03) | ‡ |
| Male | 612 (66.2) | 0.06 (0.05–0.06) | 2.33 (2.06–2.64) |
| **Race/ethnicity** | | | |
| White | 423 (45.7) | 0.03 (0.02–0.03) | ‡ |
| Hispanic | 114 (12.3) | 0.04 (0.04–0.05) | 1.69 (1.50–1.89) |
| Black/African American | 357 (38.6) | 0.15 (0.14–0.17) | 5.86 (5.30–6.49) |
| Asian/Pacific Islander [†] | 19 (2.1) | 0.02 (0.01–0.02) | 0.58 (0.47–0.70) |
| American Indian/Alaskan Native [†] | 12 (1.3) | 0.07 (0.03–0.12) | 2.42 (2.18–2.68) |
| | Frequency n (%) * | Age-Specific Mortality Rate (95% CI) | Age-Adjusted Rate Ratio (95% CI) |
| **Age (Years)** | | | |
| 0 | 41 (4.4) | 0.18 (0.12–0.23) | - |
| 1–34 | 44 (4.8) | 0.01 (0.00–0.01) | - |
| 35–44 | 49 (5.3) | 0.02 (0.01–0.03) | - |
| 45–54 | 94 (10.2) | 0.04 (0.03–0.05) | - |
| 55–64 | 197 (21.3) | 0.08 (0.07–0.09) | - |
| 65–74 | 192 (20.8) | 0.11 (0.91–0.12) | - |
| 75–84 | 162 (17.5) | 0.18 (0.15–0.21) | - |
| ≥85 | 146 (15.8) | 0.38 (0.31–0.44) | - |

95% CI = 95% confidence interval. Diagnosis based on ICD 10 Classification: Congenital Syphilis (A50), Early Syphilis (A51), Late Syphilis (A52), Other/Unspecified Syphilis (A53).  * Numbers may not add up to 100% because of rounding.  [†] Rates unreliable due to small sample size.  [‡] Referent group.

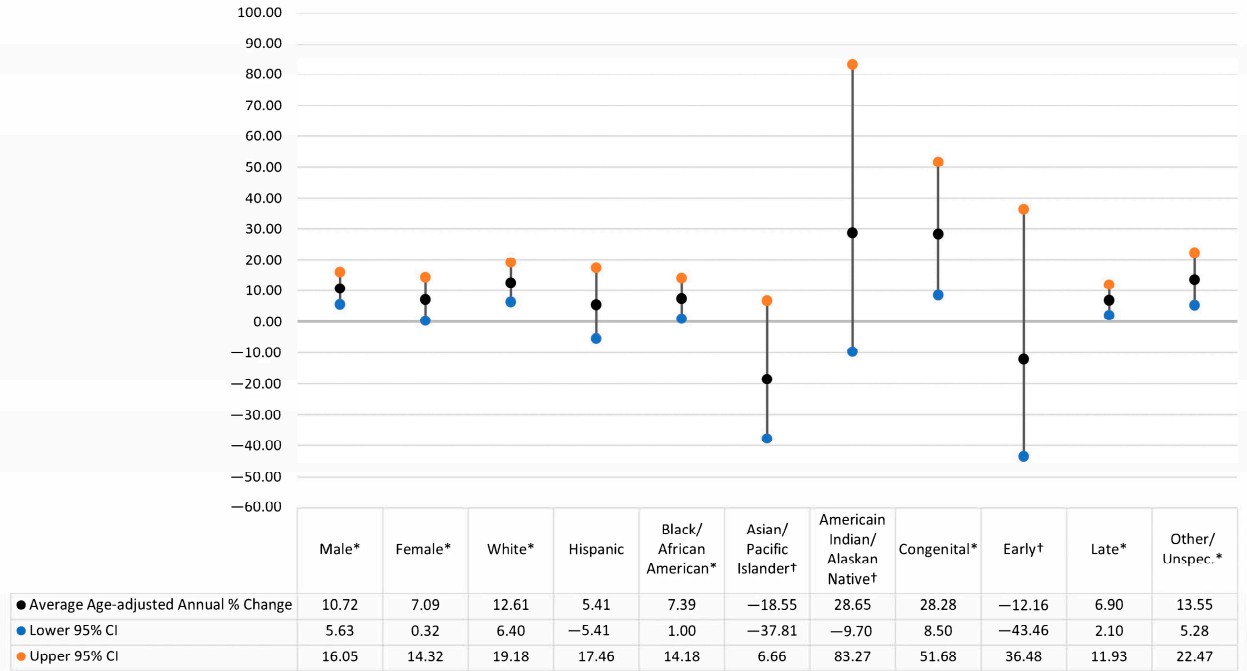

| | Male* | Female* | White* | Hispanic | Black/ African American* | Asian/ Pacific Islander† | Americain Indian/ Alaskan Native† | Congenital* | Early† | Late* | Other/ Unspec.* |
|---|---|---|---|---|---|---|---|---|---|---|---|
| ● Average Age-adjusted Annual % Change | 10.72 | 7.09 | 12.61 | 5.41 | 7.39 | −18.55 | 28.65 | 28.28 | −12.16 | 6.90 | 13.55 |
| ● Lower 95% CI | 5.63 | 0.32 | 6.40 | −5.41 | 1.00 | −37.81 | −9.70 | 8.50 | −43.46 | 2.10 | 5.28 |
| ● Upper 95% CI | 16.05 | 14.32 | 19.18 | 17.46 | 14.18 | 6.66 | 83.27 | 51.68 | 36.48 | 11.93 | 22.47 |

**Figure 2.** Age-adjusted average annual percent change in syphilis mortality rates by sex, race, and diagnosis in the United States 2015. 95% CI = 95% Confidence Interval; Diagnosis based on ICD 10 Classification: Congenital Syphilis (A50), Early Syphilis (A51), Late Syphilis (A52), Other/Unspecified Syphilis (A53). * Statistically significant ($p < 0.05$). † Rates unreliable due to small sample size.

## 4. Discussion

Similar to 2000–2014, the present analysis shows that males, Blacks/African Americans, and those aged 85 years and older continue to experience the highest burden of syphilis mortality [5]. However, in contrast to declining trends from 2000–2014, syphilis-associated mortality rates during 2015–2020 increased dramatically, with several subgroups experiencing disproportional acceleration. Among those groups experiencing the steepest increases in mortality were Whites, men, and those with congenital syphilis. The increase in congenital syphilis mortality is of particular concern due to its preventable nature with timely treatment of maternal syphilis during pregnancy [8].

It should be noted that the present analysis includes data from 2020, which marks the onset of the coronavirus disease 2019 (COVID-19) pandemic in the United States. Approximately 11% (n = 20) of syphilis-related deaths in 2020 listed COVID-19 as the underlying cause of death. Importantly, a shift in resources required for the pandemic response has been noted to have affected sexually transmitted infection programming, including disruptions in access to critical clinical services, such as testing and treatment medications [9].

This analysis is subject to a number of limitations. Because Multiple Cause of Death data are derived from death certificates, there is no way to verify the diagnosis. Furthermore, the inclusion of diagnoses on the death certificate is dependent on the certifying physician's determination that the condition contributed to death and that the physician is aware of the diagnosis. Notably, syphilis can be challenging to diagnose due to limitations of available tests and its variable, sometimes asymptomatic presentation [10]. Additionally, race and ethnicity on the death certificate are provided by an informant or in some cases on the basis of observation [6]. Finally, death certificates only provide a limited amount of demographic information in addition to the cause of death, precluding the ability to examine additional trends associated with the known risk factors of dying from syphilis.

## 5. Conclusions

Increasing syphilis mortality rates during 2015–2020 underscore the need to more systematically and comprehensively address the growing sexually transmitted infection epidemic in the US. In particular, the rapid increase in mortality from congenital syphilis is concerning, pointing to an urgent need to address the underlying drivers of disease spread, including poverty, lack of education, disparities in health care access, and other social determinants of health [1,7]. At the federal level, there is recognition that the increasing burden of syphilis morbidity and mortality must be addressed through a variety of approaches that target the individual, community, and structural factors and inequities that contribute to disease spread and progression [1]. Specifically, the Sexually Transmitted Infections National Strategic Plan for the United States, 2021–2025 outlines five key opportunities for action to curtail rising rates of sexually transmitted infections: (1) address stigma and the social determinants of health; (2) increase provider education, awareness, and training for all types of health care providers throughout the stages of their career; (3) strengthen infrastructure to better support disease surveillance and increase access to prevention and treatment services; (4) increase investment in research and innovative new vaccines, rapid diagnostics, and therapeutic and preventive antimicrobials; and (5) address concurrent health-related problems, such as the opioid epidemic, which are known to interact synergistically with and contribute to excess sexually transmitted infection burden in the community. It is hoped that through a deliberate and coordinated effort, as suggested by the National Strategic Plan, cases of syphilis and other sexually transmitted infections will begin to decline, and rates of syphilis mortality will resume a downward trend.

**Author Contributions:** Conceptualization: N.C.B., R.N.W., F.S., L.V.S. and T.K.; Methodology: F.S.; Formal Analysis: R.N.W. and N.C.B.; Writing—Original Draft Preparation: N.C.B.; Writing—Review and Editing: R.N.W., F.S., L.V.S. and T.K. All authors have read and agreed to the published version of the manuscript.

**Funding:** This research received no external funding.

**Institutional Review Board Statement:** Not applicable.

**Informed Consent Statement:** Not applicable.

**Data Availability Statement:** Publicly available data were analyzed for this study. The data can be found at https://wonder.cdc.gov/mcd.html (accessed on 7 October 2022).

**Conflicts of Interest:** The authors have no relevant financial relationship or other conflict of interest to disclose pertaining to the subject matter.

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
