# Peer review of "Current Trends in Syphilis Mortality in the United States, 2015–2020"

_venereology, doi:10.3390/venereology2020005_

Round 1

Reviewer 1 Report

Methods/ResultsRates and ratios were compared with the 2000 US population - there is no justification for using such old data. Need to use 2020 data, or - at the least, 2010.  This is a major issue that must be revised

Background: More needs to be said about existing trends and what is known, particularly in regards to mortality

Results: Consider using colour for figures - they are hard to decipher with the grey.

Discussion: Far more discussion needs to be given as to what can be done, and why these disparities on race exist. Considering comparing with other data on STIs in the US and/or other countries. 

Author Response

We would like to thank the reviewer for the comments regarding our manuscript. We are pleased to resubmit a revised version for your consideration. We think your feedback has greatly strengthened the paper. Below is a detailed table summarizing your comments and our responses to them. Attached is the full response to reviewer comments that we are uploading for the Editors, as well as the revised manuscript with tracked changes. Thanks again.

Reviewer comment and/or feedback

Changes made or author response

Reviewer 1

Methods/ResultsRates and ratios were compared with the 2000 US population - there is no justification for using such old data. Need to use 2020 data, or - at the least, 2010. This is a major issue that must be revised

The reviewer brings up an important point. We are aware of the availability of more recent standards. However, the use of the 2000 U.S. standard population was selected deliberately to align with the approach used by the United States government for annual mandated reporting on health statistics (see website for more information). We acknowledge that our choice of standard population has implications for racial and ethnic differences in mortality but feel that the need for comparability with other national data outweighs the reduction in precision. We have added language to the manuscript to clarify our use of the 2000 U.S. standard population.  

Background: More needs to be said about existing trends and what is known, particularly in regards to mortality

We thank the reviewer for this feedback. Additional information about mortality trends has been added to the introduction.

Results: Consider using colour for figures - they are hard to decipher with the grey.

We thank the reviewer for this suggestion. We have updated the figures accordingly.

Discussion: Far more discussion needs to be given as to what can be done, and why these disparities on race exist. Considering comparing with other data on STIs in the US and/or other countries. 

We thank the reviewer for this feedback. The conclusion section has been augmented to further discuss what can be done. Additionally, the introduction has been expanded to include more information about potential drivers of disparities and a comparison to other STI data.

Reviewer 2 Report

Dear Authors,

A concise report on syphilis-associated mortality in the United States, 2015-2020. It shows an alarming increase in the number of cases of preventable infection.

Could the authors indicate the importance of asymptomatic cases? These are often a great challenge because no serological and molecular tests allow timely pathogen detection (with sensitivity and specificity greater than 95%).

The authors could also indicate if Syphilis is an infection whose reporting is mandatory or not in the United States. Commonly, this infection is not reported in an official database of the health authorities of all countries.

Author Response

We would like to thank the reviewer for the comments regarding our manuscript. We are pleased to resubmit a revised version for your consideration. We think your feedback has greatly strengthened the paper. Below is a detailed table summarizing your comments and our responses to them. Attached is the full response to reviewer comments that we are uploading for the Editors, as well as the revised manuscript with tracked changes. Thanks again.

Reviewer comment and/or feedback

Changes made or author response

Reviewer 2

A concise report on syphilis-associated mortality in the United States, 2015-2020. It shows an alarming increase in the number of cases of preventable infection.

We thank the reviewer for this critical review and feedback. See below for details on how we addressed your suggested edits and questions. We believe doing so has greatly strengthened the manuscript.

Could the authors indicate the importance of asymptomatic cases? These are often a great challenge because no serological and molecular tests allow timely pathogen detection (with sensitivity and specificity greater than 95%).

We thank the reviewer for this suggestion. A comment about the importance of asymptomatic cases has been added to the Limitations section.

The authors could also indicate if Syphilis is an infection whose reporting is mandatory or not in the United States. Commonly, this infection is not reported in an official database of the health authorities of all countries.

We thank the reviewer for this suggestion. Syphilis is a mandatory reportable disease for every state within the United States. We have added this additional context to the body of the text.